# TIVC: An Efficient Local Search Algorithm for Minimum Vertex Cover in Large Graphs

**DOI:** 10.3390/s23187831

**Published:** 2023-09-12

**Authors:** Yu Zhang, Shengzhi Wang, Chanjuan Liu, Enqiang Zhu

**Affiliations:** 1Cyberspace Institute of Advanced Technology, Guangzhou University, Guangzhou 510006, China; zhangyu@e.gzhu.edu.cn; 2Institute of Computing Science and Technology, Guangzhou University, Guangzhou 510006, China; wallace_sz@163.com; 3School of Computer Science and Technology, Dalian University of Technology, Dalian 116024, China

**Keywords:** minimum vertex cover (MVC), local search, wireless sensor networks (WSNs), combinatorial optimization, large graphs

## Abstract

The minimum vertex cover (MVC) problem is a canonical **NP**-hard combinatorial optimization problem aiming to find the smallest set of vertices such that every edge has at least one endpoint in the set. This problem has extensive applications in cybersecurity, scheduling, and monitoring link failures in wireless sensor networks (WSNs). Numerous local search algorithms have been proposed to obtain “good” vertex coverage. However, due to the **NP**-hard nature, it is challenging to efficiently solve the MVC problem, especially on large graphs. In this paper, we propose an efficient local search algorithm for MVC called TIVC, which is based on two main ideas: a 3-improvements (TI) framework with a tiny perturbation and edge selection strategy. We conducted experiments on real-world large instances of a massive graph benchmark. Compared with three state-of-the-art MVC algorithms, TIVC shows superior performance in accuracy and possesses a remarkable ability to identify significantly smaller vertex covers on many graphs.

## 1. Introduction

Given an undirected graph G=(V,E), a *vertex cover* (VC) C⊆V of *G* is a subset of vertices such that every edge e∈E has at least one endpoint belonging to *C*. The *minimum vertex cover* (MVC) problem is to find a VC with the smallest size in a graph, which is a classical **NP**-hard problem with an approximation factor of 1.3606 [1]. MVC plays an important role in graph theory for its extensive applications, including scheduling [2], cybersecurity [3], and wireless sensor networks (WSNs) [4]. For example, a WSN can be modeled as an undirected graph for which vertices and edges represent infrastructures and communication links, respectively. Then, elements in VCs can be used for various purposes such as monitoring link failures, facility location, clustering, and data aggregation since each communication link (edge) is incident with at least one vertex in a VC. Figure 1 shows an example of simulating a wireless sensor network using a unit disk graph [5,6], where each vertex is the center of a circle and there is an edge between it and the other vertices within its radius. Another more specific example involves the installation of electronic cameras for a city road network to ensure that each road segment is monitored by at least one camera, thereby observing the traffic conditions. To minimize costs, it is necessary to deploy the fewest possible electronic cameras, which is equivalent to finding an MVC in the graph representing the city road network.

### 1.1. Background

The MVC problem has been extensively studied, and many algorithms have been proposed, including exact and approximate algorithms. Regarding exact algorithms, branch-and-reduce methods currently have the best time complexity [7,8]. However, the exact algorithms are still exponential-time, which cannot solve the MVC problem in a reasonable time, especially on large graphs. Therefore, approximate algorithms are proposed to solve MVC. Greedy algorithms are the common method used for approximately solving intractable problems, such as connected dominating sets [9], weighted vertex covers [10], and independent sets [11]. While greedy algorithms can quickly produce feasible solutions, the quality of solutions is generally not high enough to meet real-world requirements.

In practice, tackling intractable problems often resorts to heuristic approaches for obtaining a high-quality solution within a reasonable time, and a number of such algorithms have been proposed to address various problems, such as job shop scheduling [12], partition coloring [13], and the critical nodes problem [14]. Local search is one of the extensively studied heuristics for solving **NP**-hard problems [15,16,17,18,19]. Regarding the MVC problem, it has been shown that local search outperforms other heuristics [20]. The primary idea of local search algorithms for solving graph theory problems can be described as follows: initiate with a feasible solution and iteratively update it by removing, adding, or swapping vertices until a cutoff time is reached. A common strategy is (j,k)-swaps, i.e., removing *j* vertices from a solution and adding *k* vertices to it. We refer to a (j,k)-swap as a *j*-improvement [21]. The local search algorithms have the advantages of simple implementation and effective performance. However, they do suffer from a few challenges: the cycling phenomenon (i.e., revisiting recently visited vertices) [22,23] leads to the algorithm wasting too much computational time, resulting in a local optimum; moreover, complex vertex selection strategies may diminish the efficiency of the local search, resulting in poor performance on large graphs. To address these issues, researchers have proposed many strategies, which will be described in detail in Section 3.

### 1.2. Our Contributions

This paper proposes an efficient algorithm named TIVC for the MVC problem on large graphs. TIVC involves two main ideas. The first one is a 3-improvements framework with a tiny perturbation, which has a chance to directly search for a solution of size (k−2) after adding one vertex based on a *k*-sized feasible solution. Moreover, we use an effective edge selection strategy to accelerate search speed, which combines the *edge-age-based best from multiple selections* (EABMS) technique [20] with a random vertex selection method to choose the uncovered edges to be covered. We conduct experiments to compare TIVC with state-of-the-art local search algorithms for MVC on the Network Repository benchmark, including 72 real-world large instances. TIVC shows the best accuracy performance and significantly outperforms other algorithms in many instances.

### 1.3. Organization of This Paper

The remainder of this paper is organized as follows. Section 2 presents basic definitions. Section 3 gives a brief review of the related work on MVC. Section 4 describes the TIVC algorithm. Section 5 is devoted to the design and analysis of experiments, and Section 6 provides concluding remarks.

## 2. Preliminaries

This section introduces some preliminary knowledge. Specifically, Section 2.1 describes some notations and terminologies, and Section 2.2 briefly summarizes local search.

### 2.1. Notations and Terminologies

Denote by G=(V,E) an undirected graph with *vertex set V* and *edge set E*. For an edge e=(u,v), the two vertices *u* and *v* are called *endpoints* of *e*. A vertex is *adjacent* to another vertex if they are the two endpoints of an edge, and one is called a *neighbor* of the other. An edge is *incident* with each of its endpoints. The set consisting of all neighbors of a vertex v∈V, denoted by N(v), is the *neighborhood* of *v*, and N[v]=N(v)∪{v} is the *closed neighborhood* of *v*. The degree of *v* is the number of edges incident with *v*. For a vertex set S⊆V, let $N\left[S\right]={⋃}_{v\in S}N\left[v\right]$ and N(S)=N[S]∖S.

For a graph G=(V,E) and a set of vertices S⊆V, an edge e∈E is *covered* by *S* if at least one endpoint of *e* belongs to *S*; otherwise, *e* is *uncovered* by *S*. If all edges of *G* are covered by *S*, then *S* is called a *vertex cover* (VC) of *G*. A VC with the smallest cardinality is called a *minimum vertex cover* (MVC) of *G*. Note that a graph G=(V,E) may have more than one MVC. We use $E_{u}\left(S\right)\subseteq E$ to denote the set of edges uncovered by *S* and use $E_{c}\left(S\right)\subseteq E$ to denote the set of edges covered by *S*. The MVC problem is to find an MVC from a graph.

### 2.2. Local Search

From this section, *C* represents a candidate (or partial) solution of the MVC problem. The general scheme of local search for MVC is to construct an initial VC first and then iteratively improve the solution to a smaller one by vertex swapping. Generally, local search algorithms use *gain(v)* and *loss(v)* to measure the importance of a vertex *v*, where *gain(v)* denotes the number of edges uncovered by *C* but covered by C∪{v}, and *loss(v)* is the number of edges covered by *C* but uncovered by C∖{v}. The *age* of a vertex *v*, denoted by *age(v)*, is the number of steps since it was last removed from *C*. The *age* values are usually used to break ties, where ties mean the existence of multiple vertices with the same *gain* or *loss*. In addition, the age of an edge *e*, denoted by *age(e)*, is the number of steps since it was last uncovered by *C*, which is often used as a criterion for selecting edges [20].

## 3. Related Work

This section provides a brief review on heuristic algorithms for MVC. In 2013, Cat et al. [24] proposed a two-stage strategy that allows the selection of a pair of vertices separately and exchanges vertices in two stages, based on which a NuMVC algorithm for MVC is developed, which addresses the drawback (time-consuming) of previous algorithms that require selecting vertices simultaneously [22,25,26]. However, with the rapid development of the Internet and the widespread deployment of sensors, the size of datasets has dramatically increased, and many algorithms fail to solve MVC on large instances. For this, Cai et al. [27] introduced the *Best from Multiple Selections* (BMS) heuristic, which randomly samples *k* vertices in *C* and removes one with the minimum *loss* value from *C*. This heuristic aims to obtain a trade-off between efficiency and accuracy. Based on BMS, an algorithm named FastVC is developed for solving MVC well on large instances. By combining BMS and the best-picking strategy [24], Ma et al. [28] proposed best-picking with a noisy strategy and developed an algorithm NoiseVC; they also proposed a *BMS with random walk strategy* (WalkBMS) in another study [29], which selects (with a probability) BMS or random walk as the vertex selection strategy to handle the issue with FastVC becoming easily trapped in a local optimum. Subsequently, Cai et al. [30] proposed an improved version of FatVC, named FastVC2+p, by integrating some processing techniques and initial solution construction methods. In 2019, Luo et al. [31] proposed a highly parametric local search framework for MVC, called MetaVC, which incorporates many effective local search techniques. In addition, the authors used an automatic algorithm configurator that sets parameters for the type of instances to maximize the performance of MetaVC. In 2021, Quan et al. [20] proposed a new edge weighting method based on edge age (EABMS), which randomly samples *a* edges in $E_{u}\left(C\right)$ and selects one edge with the maximum *age* value for covering (by adding one of its endpoints to *C*). Based on EABMS, an algorithm EAVC and its variant EAVC2+p have been developed for MVC. Both EAVC and EAVC2+p showed superiority on large graphs compared with FastVC and its variants. To date, FastVC, MetaVC, EAVC, and their variants are state-of-the-art MVC local search algorithms for large instances. To demonstrate the effectiveness of TIVC, we compared our algorithm with the baseline algorithms, i.e., FastVC, MetaVC, and EAVC.

## 4. Main Algorithm

In this section, we describe our algorithm TIVC. We first introduce the top-level architecture of TIVC and then describe the algorithm in detail. Finally, we give a complexity analysis for TIVC. Note that in this section, *C* represents a candidate (or partial) solution.

### 4.1. Top-Level Architecture

The top-level architecture of TIVC is shown in Algorithm 1. TIVC starts with constructing an initial VC *C* for the graph *G* (line 1) and then enters a loop for finding a VC as small as possible within a given cutoff time (lines 2–10). Specifically, when obtaining a VC *C*, it updates the best solution $C^{*}$ and then removes a vertex from *C* (lines 3–6). If *C* is not a feasible solution, then the algorithm iteratively exchanges vertices until *C* becomes a VC. First, it removes vertices from *C* until $\left|C|=\right|C^{*}|-3$ (line 7). Next, it selects an uncovered edge and adds one of its endpoints to *C* (line 8). If *C* remains infeasible, it selects another vertex to add to *C* in the same way (lines 9–10). Finally, the best-found vertex cover $C^{*}$ is returned when the cutoff time is reached (line 11).
**Algorithm 1:** Top Level of TIVC **Input:** A graph  G = (V,E), the *cutoff* time **Output:** A vertex cover C* of G
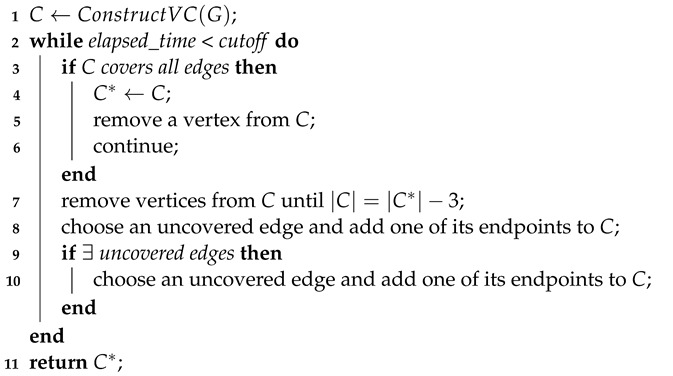


### 4.2. The TIVC Algorithm

Our TIVC algorithm is shown in Algorithm 2, which encompasses two stages, i.e., construction and search.
**Algorithm 2:** TIVC **Input:** A graph  G = (V,E), the *cutoff* time **Output:** A vertex cover C* of G
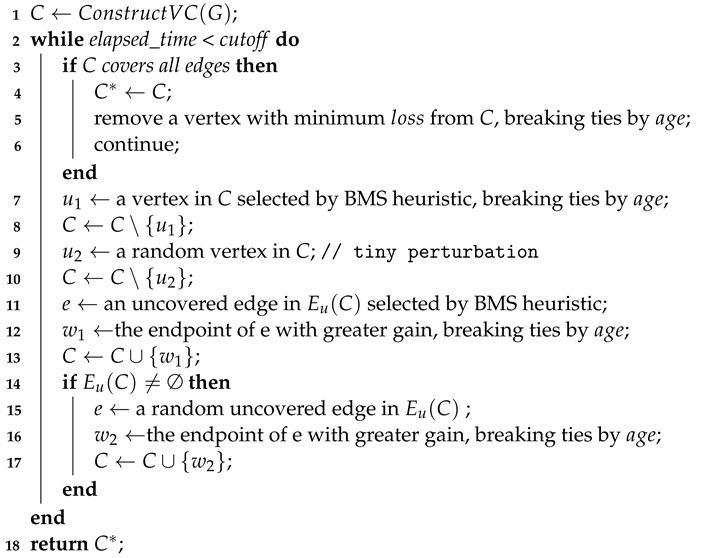


In the construction stage, the algorithm constructs an initial VC *C* of *G* (line 1) by EdgeGreedyVC [20,27,31], which is a commonly used approach for MVC algorithms. The process starts with an empty set *C* and proceeds iteratively by checking and covering edges to extend *C*. Once a VC is obtained, redundant vertices are removed from *C*, where redundant vertices are those with *loss = 0* and removing them does not produce new uncovered edges.

In the search phase, the algorithm attempts to iteratively remove vertices from *C* and add vertices to *C* to search for a VC smaller than the current best solution $C^{*}$. First, it repeatedly removes a vertex with the minimum *loss* from *C* until *C* is not a VC, i.e., $\left|C|=\right|C^{*}|-1$ (lines 3–6). Second, the algorithm repeatedly performs vertex swapping. Each swapping step contains a removing phase (lines 7–11) and an adding phase (lines 12–17). In the removing phase, the first vertex $u_{1}$ is selected by the BMS heuristic and removed from *C* (lines 7–8); then, the second vertex is selected randomly in *C* and removed from *C* to perturb the solution slightly (lines 9–10). The above implementation lead to $\left|C|=\right|C^{*}|-3$. In the adding phase, it first selects an uncovered edge $e\in E_{u}\left(C\right)$ and adds the vertex with greater *gain* in its endpoints to *C* (lines 11–13); if there are uncovered edges ($E_{u}\left(C\right)\neq \emptyset$), then it chooses one edge in $E_{u}\left(C\right)$ randomly and adds the vertex with greater *gain* to *C* (lines 14–17). If *C* is a feasible VC at this time, then the algorithm completes a 3-improvement; otherwise, the algorithm continues to perform lines 7–17. Finally, the best-found VC $C^{*}$ is returned when the cutoff time is reached (line 18).

An important implementation detail is that when a vertex *v* is removed from or added to *C*, the gain or loss of vertices in N[v] needs to be updated accordingly.

### 4.3. Complexity Analysis

In this section, we analyze the time complexity of TIVC. For a given graph G=(V,E), let |V|=n,|E|=m.

**Proof.** TIVC (Algorithm 2) runs in O(m+n). First, the *ConstructVC* procedure (line 1) constructs an initial solution by EdgeGreedyVC, which has a time complexity of O(m) [27]. Second, lines 3–6 take O(m+n) time since the procedure traverses *E* and *V* once. Third, lines 7–17 take O(m) time because the time complexity of BMS and EABMS has already been proven to be O(1) [20,27]; the time complexity of removing a vertex from a solution *C* and adding a vertex to *C* is O(1), and the time complexity of updating gain or loss is O(Δ), where Δ is the maximum degree of the graph; Line 14 needs to traverse *E* once for checking the condition of whether $E_{u}\left(C\right)\neq \emptyset$. Thus, the time complexity of TIVC is O(m+n). □

## 5. Results and Discussion

In this section, we evaluate TIVC on the Network Repository benchmark (https://networkrepository.com accessed on 9 March 2023) [32]. This benchmark includes enormous amounts of graphs from various areas. To assess the performance of TIVC on large graphs, we specifically selected instances with vertex numbers ranging from $10^{4}$ to $10^{7}$, encompassing 72 instances. Section 5.1 introduces the design of the experiment; Section 5.2 reports the results of the experiment; Section 5.3 discusses and analyzes the experimental results.

### 5.1. Experiment Setup

TIVC is implemented in C++ and compiled by gcc 7.1.0 with the `-O3’ optimization option. All experiments are run under CentOS Linux release 7.6.1810 with an Intel(R) Xeon(R) Gold 6254 CPU@3.10GHz with 128 GB RAM. The parameters of FastVC and EAVC are set to be the same as those used in the original literature [20,27], and the parameters of MetaVC are set according to the recommended values in reference [31] on large instances (REAL-WORLD). However, no processing techniques are employed, aligning with the other algorithms. TIVC incorporates two tunable parameters: *k* for the BMS and *a* for the EABMS. These parameters are set to 50 and 24, respectively, aligning with the settings of EAVC.

We compare TIVC with three state-of-the-art local search algorithms: FastVC [27], MetaVC [31], and EAVC [20] for MVC. All three algorithms are suitable for solving large instances for MVC. FastVC combines the two-stage exchange framework and the BMS heuristic to balance the algorithm’s accuracy and efficiency, which achieves good performance on large graphs. MetaVC integrates many local search techniques based on the two-stage framework and incorporates an automatic configurator to select and set parameters. For large instances, MetaVC involves BMS, reconstruction, and random walk mechanisms, where reconstruction means removing *t* vertices from the solution *C* during the search phase and then adding *t* vertices with the greatest gains in V∖C to *C*. EAVC is based on the two-stage framework and combines WalkBMS and EABMS to provide good guidance for improving the quality of solutions (and also increasing the diversity of solutions) in the vertex and edge selection phase.

Table 1 shows the details of these four algorithms. The construction procedures of the four algorithms are based on EdgeGreedyVC. For vertex selection, FastVC, MetaVC, EAVC, and TIVC utilize BMS, BMS + Random, WalkBMS, and BMS + Random strategies, respectively. Regarding edge selection, FastVC, MetaVC, EAVC, and TIVC use Random, Random, EABMS, and EABMS+Random strategies, respectively. In addition, FastVC, MetaVC, and EAVC are both based on 2-improvements, while TIVC is based on 3-improvements. Note that MetaVC also incorporates a reconstruction mechanism during the search phase. The codes of FastVC (http://lcs.ios.ac.cn/~caisw/VC.html accessed on 8 July 2023), MetaVC (https://github.com/chuanluocs/MetaVC accessed on 18 August 2023), and EAVC (https://github.com/quancs/EAVC accessed on 8 July 2023), are open online.

For each instance, all algorithms are executed 10 times with seeds 1, 2, 3, …, 10. The cutoff time for each run is set at 1000 s. For each instance, we present the best (i.e., smallest) solution as Min, the average solution as Avg, and the average running time (over the 10 runs) as $t_{avg}$.

### 5.2. Experimental Result

Results on the Network Repository benchmark are reported in Table 2. TIVC shows superior performance in terms of accuracy in the majority of instances, outperforming both the FastVC, MetaVC, and EAVC algorithms. Specifically, TIVC obtains the best solution on 50 (out of 72) instances, while FastVC, MetaVC, and EAVC obtain 26, 28, and 37 best solutions, respectively. In particular, TIVC shows a remarkable ability to find strictly optimal solutions—in total, 20 such solutions. In comparison, FastVC, MetaVC, and EAVC can find 3, 12, and 7 strictly optimal solutions, respectively. Regarding the average solution, TIVC also outperforms the other algorithms. FastVC, MetaVC, EAVC, and TIVC obtain the optimal average VC on 7, 12, 13, and 20 instances, respectively. In addition, for large instances with $10^{7}$ vertices, TIVC performs remarkably well and finds much smaller VCs than other algorithms on many instances.

Moreover, we report summary results for each algorithm on instances with orders (the number of vertices) from $10^{4}$ to $10^{7}$, as shown in Table 3. There are 7, 26, 27, and 12 instances whose orders are $10^{4},10^{5},10^{6},$ and $10^{7}$, respectively. All algorithms found the best solutions on the instances of order $10^{4}$. Regarding the instances of order $10^{5}$, the numbers of instances on which TIVC, FastVC, MetaVC, and EAVC found the best solutions (strictly optimal solutions) are 18, 7, 10, and 12 (8, 0, 6, 2), respectively. For the instances of order $10^{6}$, TIVC and EAVC had similar performance—finding the best solutions on 17 instances and the strictly optimal solutions on 5 instances—while FastVC and EAVC found the best solutions on 10 and 9 instances, respectively, and obtained strictly optimal solutions on 1 and 4 instances. Finally, regarding the instances of order $10^{7}$, the numbers of instances on which TIVC, FastVC, MetaVC, and EAVC found the best solutions (strictly optimal solutions) are 8, 2, 2, and 1 (7, 2, 2, 0), respectively.

As mentioned in Section 1, MVC has a real-world application in electronic camera installation on road networks. To capture visual information on a road network, such as traffic conditions, vehicle positions and speeds, and pedestrian flows, we need to install cameras at the intersections of roads, guaranteeing that every road can be monitored by at least one camera. Observe that a camera can monitor more than one road. In practice, to save costs, it is sufficient to install cameras at a small number of intersections. Now, a problem arises: given a road network, what are the intersections for installing cameras such that the number of cameras is minimized and all roads are monitored? By modeling a road network as a graph whose vertices represent intersections and for which two vertices are connected by an edge if and only if they are connected by a road (without considering the length and width of roads), this problem is equivalent to finding an MVC in a graph. As an example, we consider the instance “inf-road-usa”, which is a graph abstracted from the road network in the United States. As shown in Table 2, the solution for this instance provided by our TIVC algorihtm is 11,950,231, outperforming the suboptimal solution 11,989,552; i.e., compared with other state-of-the-art algorithms, TIVC can save at least 39,321 cameras when installing cameras for monitoring this network.

### 5.3. Discussion

The experimental results demonstrate the effectiveness of our TIVC algorithm for solving MVC on large graphs. From Section 5.2, it can be seen that TIVC fails to find the best solution mainly on instances of orders $10^{5}$ and $10^{6}$. A possible explanation for this might be that the tiny perturbation guidance algorithm focuses mainly on the solution diversity when falling into a local optimum, ignoring the intensity of the improvement of the current solution. Nevertheless, the gap between a solution returned by TIVC and the best solution is small, generally within $10^{2}$. In comparison, on instances of order $10^{7}$, algorithms in our experiments are less likely to become trapped in local optima due to the large solution space (the t_avg values of all four algorithms are close to 1000 s). In this case, the advantage of the 3-improvements framework is showcased prominently, i.e., it has a chance to directly search for a solution of size (k−2) after adding one vertex based on a *k*-sized feasible solution, whereas 2-improvements cannot achieve this.

TIVC has certain limitations. First, for small instances, the diversity of the solution space may be limited and the perturbation could potentially lead the search towards a suboptimal direction, which hinders TIVC from finding the best solution within the time threshold. In contrast, probabilistic (rather than fixed) perturbations are more likely to guide the algorithm into an optimal direction. Nevertheless, the difference between the solutions found by TIVC and the best solutions obtained by other algorithms are not significant (e.g., the instances “wave”, “rec-dating”, “citationCitesee”, “web-Stanford”, etc.). In addition, all algorithms considered in this paper except MetaVC have a common limitation, i.e., parameter setting. For different types of instances, different parameter settings may affect the performance of the algorithms to a certain extent. MetaVC has automatic parameter configurations that can tune parameters automatically according to the types of instances; this flexibility could be a contributing factor to its outstanding performance on some instances, such as “hugetrace-00020” and “hugebubbles-00000”.

## 6. Conclusions

In this paper, we propose an efficient local search algorithm for the MVC problem called TIVC, which consists of a 3-improvements framework with a tiny perturbation and edge selection strategy. The experimental results show that TIVC significantly outperforms state-of-the-art algorithms for MVC on large graphs, especially those with orders exceeding $10^{7}$ (although such instances are scarce currently, they will become increasingly prevalent with the development of the internet and big data). This provides an enhanced approach for designing and analyzing large WSNs.

In the future, we aim to explore more efficient approaches for solving various graph theoretical problems on large-scale graphs—especially algorithms with theoretical guarantees. Additionally, we plan to integrate local search with advanced machine learning methods to accelerate the convergence speed of local search algorithms, allowing them to autonomously terminate the search process instead of relying solely on time constraints. Finally, exploring more real-world problems that can be modeled to the MVC-related problems (and applying our algorithm to solve them) is also further work we will consider.

## Figures and Tables

**Figure 1 sensors-23-07831-f001:**
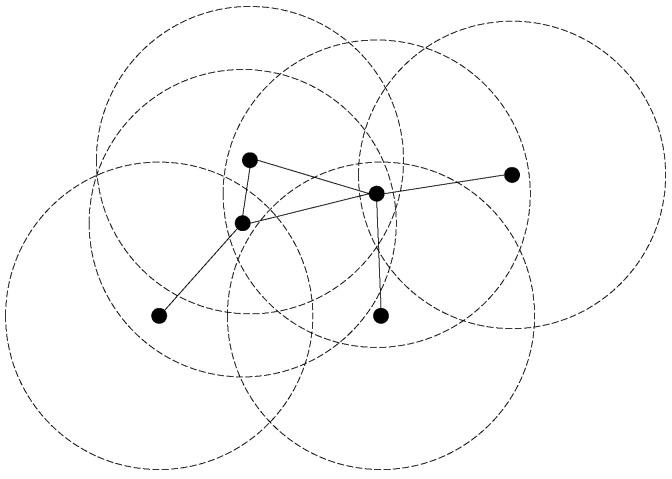
An example of a WSN.

**Table 1 sensors-23-07831-t001:** Equipment of four algorithms.

Algorithms	FastVC	MetaVC	EAVC	TIVC
Construction	EdgeGreedyVC	EdgeGreedyVC	EdgeGreedyVC	EdgeGreedyVC
Vertex Selection	BMS	BMS + Random	WalkBMS	BMS + Random
Edge Selection	Random	Random	EABMS	EABMS + Random
k-improvement	2	2	2	3

**Table 2 sensors-23-07831-t002:** The results on the Network Repository benchmark.

Instance	FastVC			MetaVC			EAVC			TIVC		
	Min	Avg	t_avg	Min	Avg	t_avg	Min	Avg	t_avg	Min	Avg	t_avg
cit-HepTh	**18,155**	18,155.0	0.05	**18,155**	18,155.0	0.40	**18,155**	18,155.0	0.12	**18,155**	18,155.4	0.12
as-22july06	**3303**	3303.0	0.01	**3303**	3303.0	0.04	**3303**	3303.0	2.41	**3303**	3303.6	0.01
cit-HepPh	**22,589**	22,589.0	0.34	**22,589**	22,589.0	1.89	**22,589**	22,589.0	0.21	**22,589**	22,589.2	0.37
cond-mat-2005	**23,106**	23,106.0	0.09	**23,106**	23,106.0	0.33	**23,106**	23,106.0	0.05	**23,106**	23,106.4	0.07
soc-Epinions1	**22,280**	22,280.0	0.14	**22,280**	22,280.0	1.58	**22,280**	22,280.0	0.11	**22,280**	22,280.5	0.10
soc-Slashdot0811	**24,046**	24,046.0	0.31	**24,046**	24,046.0	2.19	**24,046**	24,046.0	0.15	**24,046**	24,046.4	0.14
soc-Slashdot0902	**25,770**	25,770.2	0.29	**25,770**	25,770.0	12.10	**25,770**	25,772.3	308.62	**25,770**	25,770.6	265.89
luxembourg_osm	**56,936**	56,937.6	186.24	56,937	56,937.0	306.05	**56,936**	56,937.9	37.14	**56,936**	56,936.5	68.12
wave	119,306	119,445.9	659.30	**119,279**	119,309.9	596.04	119,302	119,382.1	561.16	119,290	119,331.3	607.95
rec-dating	89,839	89,853.3	302.80	**89,780**	89,787.9	809.35	89,806	89,811.4	752.09	89,792	89,799.7	798.95
caidaRouterLevel	75,433	75,443.8	6.17	75,201	75,209.2	927.27	75,199	75,215.8	235.86	**75,173**	75,190.5	933.96
rec-libimseti-dir	94,198	96,275.5	479.42	93,710	96,033.1	858.66	93,704	93,708.0	560.67	**93,696**	93,701.4	742.14
coAuthorsCiteseer	**129,193**	129,193.0	1.32	**129,193**	129,193.0	26.94	**129,193**	129,193.0	0.58	**129,193**	129,193.6	0.60
amazon0302	168,554	168,557.9	574.72	168,553	168,561.1	590.24	168,547	168,552.5	467.84	**168,545**	168,550.6	556.79
email-EuAll	18,317	18,317.0	0.06	**18,316**	18,316.0	31.62	**18,316**	18,316.1	0.07	**18,316**	18,317.2	0.07
Ga41As41H72	237,101	237,985.0	997.21	233,534	233,923.0	625.24	233,550	233,604.9	636.46	**233,508**	233,632.2	397.48
citationCiteseer	118,180	118,184.6	12.27	**118,118**	118,120.2	863.87	118,135	118,147.6	84.15	118,135	118,146.0	440.38
web-Stanford	118,879	118,895.1	637.11	**118,591**	118,605.3	836.93	118,613	118,625.2	444.53	118,603	118,616.2	878.72
coAuthorsDBLP	**155,618**	155,618.0	5.94	**155,618**	155,618.0	164.00	**155,618**	155,618.0	1.66	**155,618**	155,619.0	1.13
ca-dblp-2012	**164,949**	164,949.0	3.74	**164,949**	164,949.1	387.89	**164,949**	164,949.0	1.29	**164,949**	164,949.7	0.76
cnr-2000	96,091	96,104.1	368.16	**95,700**	95,725.2	935.77	95,778	95,798.4	768.95	95,735	95,769.6	930.46
web-NotreDame	74,094	74,104.3	128.56	**73,914**	73,920.1	862.43	73,943	73,948.3	394.22	73,927	73,931.7	895.30
amazon0312	261,594	261,598.7	840.60	261,622	261,632.5	604.34	261,596	261,602.4	422.35	**261,591**	261,598.5	834.25
amazon0601	266,579	266,586.3	461.46	266,607	266,615.6	578.30	266,567	266,572.5	346.39	**266,565**	266,572.2	662.40
amazon0505	267,256	267,260.3	682.73	267,280	267,288.4	615.81	267,252	267,257.4	505.85	**267,248**	267,254.7	715.89
coPapersCiteseer	**386,106**	386,106.0	9.86	386,112	386,117.4	556.86	**386,106**	386,106.0	5.76	**386,106**	386,106.9	6.15
ca-coauthors-dblp	**472,179**	472,179.0	20.44	472,194	472,199.1	508.31	**472,179**	472,179.0	5.01	**472,179**	472,179.9	4.42
coPapersDBLP	**472,179**	472,179.0	35.57	472,194	472,197.2	333.46	**472,179**	472,179.0	8.36	**472,179**	472,179.9	8.03
web-BerkStan	278,906	278,934.2	799.02	277,400	277,448.3	966.86	277,209	277,228.0	546.69	**277,200**	277,219.2	843.83
rec-epinion	100,435	100,444.3	5.65	100,180	100,199.4	922.50	**100,011**	100,016.7	2.10	**100,011**	100,017.3	113.18
eu-2005	412,377	412,397.6	879.54	411,702	411,788.7	952.56	**411,007**	411,040.6	906.16	411,015	411,034.9	850.33
web-Google	346,920	346,924.7	131.10	346,756	346,775.3	889.98	**346,672**	346,680.8	256.54	346,673	346,680.4	283.37
ldoor	899,422	899,423.2	292.15	899,423	899,429.2	726.92	**899,420**	899,421.0	41.02	**899,420**	899,421.3	81.56
inf-roadNet-PA	555,231	555,251.8	662.74	**554,890**	554,956.9	940.83	555,260	555,316.0	379.00	555,258	555,315.9	868.15
rt-retweet-crawl	81,042	81,044.6	79.53	**81,040**	81,040.0	433.04	81,041	81,041.8	0.74	81,041	81,042.0	59.16
soc-youtube-snap	**276,945**	276,945.0	12.70	276,946	276,947.3	434.07	**276,945**	276,945.7	5.08	**276,945**	276,946.2	6.62
soc-lastfm	**78,688**	78,688.0	0.34	**78,688**	78,688.0	0.68	**78,688**	78,688.0	0.67	**78,688**	78,688.3	0.68
in-2004	487,189	487,237.8	902.55	486,920	486,953.8	936.79	**486,490**	486,519.0	926.08	486,509	486,519.0	867.07
tech-as-skitter	527,163	527,201.2	410.91	526,529	526,635.7	964.58	525,494	525,515.5	520.16	**525,492**	525,515.5	497.47
soc-flickr-und	**474,637**	474,637.5	233.62	474,646	474,652.8	579.93	**474,637**	474,637.9	94.72	**474,637**	474,638.5	69.20
inf-roadNet-CA	1,001,317	1,001,341.0	901.75	**1,001,123**	1,001,246.6	872.86	1,001,473	1,001,525.0	437.53	1,001,471	1,001,513.0	749.76
web-baidu-baike	637,106	637,110.2	506.66	637,092	637,111.9	783.86	637,014	637,019.8	330.01	**637,013**	637,021.0	384.52
packing*b050	1,624,945	1,625,325.0	996.99	**1,615,573**	1,616,094.7	976.60	1,624,191	1,625,500.0	997.79	1,623,445	1,625,416.0	997.74
tech-ip	**67,007**	67,007.0	1.35	**67,007**	67,007.0	1.77	**67,007**	67,007.0	4.92	**67,007**	67,007.4	2.91
soc-flixster	**96,317**	96,317.0	1.07	**96,317**	96,317.0	1.08	**96,317**	96,317.0	1.56	**96,317**	96,317.9	1.35
socfb-B-anon	**303,048**	303,048.9	42.41	**303,048**	303,048.4	331.14	**303,048**	303,048.2	3.17	**303,048**	303,048.7	3.21
soc-orkut	2,171,329	2,171,379.0	996.62	2,171,342	2,171,413.0	916.41	2,171,270	2,171,301.0	997.66	**2,171,213**	2,171,291.0	993.39
soc-orkut-dir	2,233,961	2,234,015.0	996.91	2,233,979	2,234,033.8	976.94	**2,233,775**	2,233,820.0	997.14	2,233,858	2,233,929.0	996.25
socfb-A-anon	375,231	375,232.8	27.41	**375,230**	375,231.8	340.28	**375,230**	375,230.9	100.17	**375,230**	375,230.9	75.81
patents	1,673,977	1,674,016.0	982.94	1,673,793	1,673,839.9	913.50	**1,673,562**	1,673,615.0	969.76	1,673,600	1,673,632.0	976.26
soc-livejournal	1,869,043	1,869,052.0	928.87	1,869,216	1,869,260.7	693.26	1,868,986	1,868,991.0	871.12	**1,868,982**	1,868,991.0	860.13
delaunay_n22	2,873,973	2,874,015.0	999.09	2,877,830	2,878,304.9	902.49	2,873,305	2,873,348.0	999.37	**2,873,207**	2,873,239.0	999.19
ljournal-2008	2,393,023	2,393,035.0	948.56	2,393,204	2,393,256.1	891.12	**2,392,664**	2,392,681.0	879.10	2,392,666	2,392,682.0	848.51
soc-ljournal-2008	2,392,992	2,393,038.0	950.13	2,393,179	2,393,265.3	861.51	**2,392,660**	2,392,677.0	780.64	2,392,665	2,392,679.0	873.16
rel9	**273,993**	273,993.4	274.61	274,155	274,160.0	337.87	**273,993**	273,993.5	152.69	**273,993**	273,993.6	214.00
sc-rel9	**273,993**	273,993.3	333.19	274,155	274,160.0	352.04	**273,993**	273,993.4	258.32	**273,993**	273,993.6	198.41
soc-livejo*groups	1,841,367	1,841,386.0	907.27	1,841,441	1,841,501.7	885.40	**1,841,061**	1,841,077.0	503.85	**1,841,061**	1,841,078.0	643.42
delaunay_n23	**5,753,835**	5,754,557.0	999.97	5,791,630	5,793,429.5	999.93	5,799,719	5,801,179.0	999.99	5,762,821	5,763,597.0	999.99
friendster	1,038,252	1,038,257.0	588.53	1,038,262	1,038,271.7	699.72	**1,038,239**	1,038,242.0	374.20	**1,038,239**	1,038,244.0	309.58
relat9	**274,297**	274,297.0	4.19	274,395	274,395.0	0.57	**274,297**	274,297.0	139.00	**274,297**	274,297.1	78.58
inf-germany_os	**5,710,522**	5,710,676.0	999.93	5,743,216	5,748,327.0	999.96	5,777,786	5,786,140.0	999.99	5,714,456	5,715,014.0	999.98
hugetrace-00010	**6,650,729**	6,754,798.0	1000.00	6,782,555	6,809,180.5	999.99	6,914,568	6,924,276.0	1000.00	6,762,321	6,764,085.0	1000.00
road_central	6,902,108	6,911,566.0	1000.00	6,898,380	6,926,199.7	999.96	6,944,280	6,945,766.0	999.99	**6,890,545**	6,895,530.0	999.99
hugetrace-00020	9,293,370	9,334,922.0	1000.00	**9,232,864**	9,421,598.0	1000.00	9,321,757	9,333,712.0	1000.00	9,239,511	9,240,936.0	1000.00
delaunay_n24	11,850,819	11,867,478.0	1000.00	11,954,634	11,958,292.5	1000.00	11,871,283	11,874,602.0	999.99	**11,823,278**	11,824,566.0	999.99
hugebubbles-00000	10,469,546	10,498,559.0	1000.00	**10,412,746**	10,551,611.5	1000.00	10,508,631	10,511,251.0	1000.00	10,417,508	10,432,354.0	1000.00
uk-2002	6,642,980	6,650,452.0	1000.00	6,662,784	6,695,299.3	1000.00	6,588,420	6,588,632.0	999.55	**6,588,132**	6,588,738.0	999.60
hugebubbles-00010	11,667,812	11,695,490.0	1000.00	11,666,025	11,718,530.0	1000.00	11,352,764	11,360,173.0	1000.00	**11,319,443**	11,328,490.0	1000.00
hugebubbles-00020	12,658,142	12,668,880.0	1000.00	12,593,050	12,695,334.2	1000.00	12,406,556	12,407,781.0	1000.00	**12,358,096**	12,359,528.0	1000.00
inf-road-usa	12,022,434	12,027,594.0	1000.00	12,024,687	12,049,621.1	1000.00	11,989,552	11,991,769.0	999.99	**11,950,231**	11,952,066.0	999.99
inf-europe_os	25,910,226	25,912,219.0	1000.00	25,911,227	25,920,918.3	999.90	25,918,018	25,924,685.0	1000.00	**25,896,775**	25,898,696.0	999.99
socfb-uci-uni	866,768	866,768.0	43.18	866,767	866,767.8	160.31	**866,766**	866,766.0	26.20	**866,766**	866,767.4	24.50

**Table 3 sensors-23-07831-t003:** Summary results. “A/B” indicates that the corresponding algorithm found A best solutions and B strictly optimal solutions in instances of the corresponding scale.

Instance_Scale	Num	FastVC	MetaV	EAVC	TIVC
$10^{4}$	7	**7**/**0**	**7**/**0**	**7**/**0**	**7**/**0**
$10^{5}$	26	7/0	10/6	12/2	**18**/**8**
$10^{6}$	27	10/1	9/4	**17**/**5**	**17**/**5**
$10^{7}$	12	2/2	2/2	1/0	**8**/**7**

## Data Availability

The data used in this study are openly available at https://networkrepository.com accessed on 9 March 2023, reference number [32].

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
