# Peer review of "TIVC: An Efficient Local Search Algorithm for Minimum Vertex Cover in Large Graphs"

_sensors, 2023, doi:10.3390/s23187831_

Round 1

Reviewer 1 Report

Results should be compared with any other recent works

Proposed method should be explain little more

No need to discuss more in introduction about your proposed work

inference of the survey should be added

Little improve

Author Response

Dear Reviewer,

We sincerely appreciate your review of our paper and your valuable comments. We revised the paper according to your comments and suggestions, where the revised parts are highlighted in red. Below is our response to your comments:

Comment 1: Results should be compared with any other recent works.

Reply: Thanks for the suggestion! Regarding the comparison with other works, we added a comparison with MetaVC (the algorithm was proposed by Luo et al. in IJCAI-19) to better highlight the advantages and innovations of our proposed method. The changes made in response to this comment involve lines 9, 122--125, 130, 133, 195--198, 201--209, 213--220, 229--234, Table 1 (page 6), Table 2 (page 8), and Table 3 (page 9).

Comment 2: Proposed method should be explained little more.

Reply: Thanks for pointing out the inadequacy of TIVC's description! We have added more description on TIVC in the revised paper, in hoping that the readers can better understand our algorithm. The changes made in response to this comment involve lines 137, 157--160, 168, 169.

Comment 3: No need to discuss more in introduction about your proposed work.

Reply: Thank you very much. Regarding the discussion in the Introduction section, following your suggestion, we reduced the description of our proposed algorithm in the Introduction. The changes made in response to this comment involve lines 60--66. In addition, we divided Section 1 into three subsections to optimize the structure of the paper.

Comment 4: Inference of the survey should be added.

Reply: Thanks for the suggestion! We apologize for any confusion. If you feel that the revised version provided below does not accurately meet your intended meaning, please provide us with further feedback. We added inferences drawn from experimental results to our Conclusions section (lines 266--269), suggesting that TIVC might exhibit more notable performance on large instances. Although instances of this scale are currently scarce, with the ongoing development of future networks, larger instances will become increasingly prevalent.

We have checked the paper again, hoping that there are no other mistakes. Thanks again for all your help.

Reviewer 2 Report

This paper proposes an efficient local search algorithm, called TIVC, for solving the minimum vertex cover (MVC) problem. MVC is a well-known challenging problem with wide applications in wireless sensor networks and has been extensively studied. The idea consists of a 3-improvements framework with a tiny perturbation and an edge selection strategy. The paper is well organized and the authors have conducted adequate experiments to verify the effectiveness of the algorithm. It is recommended that the paper be accepted with minor revisions. There are some comments that shall be considered to improve the quality of this manuscript:

1.      The authors should add a description of the unit disk graphs, explaining why they can be used to simulate WSNs.

2.      The updates of gain and loss should be mentioned in the description of TIVC and added in the complexity analysis.

3.      The WalkBMS strategy should be briefly described in Related Work.

4.      The authors mentioned an interest in working on other graph theory problems on large graphs, a brief description of the challenges that may follow can be added.

5.      The authors can briefly include in the conclusion a reference to how this work has contributed to solving these practical problems.

6.      In line 105, the word "the" needs to be inserted before "age values".

7.      The "a" should be added before "tiny perturbation" in the Abstract and Conclusions.

8.      The "age" in line 107 is different from the previous format.

Please see my comments regarding presentation issues in the section titled 'Comments and Suggestions for Authors'.

Author Response

Dear Reviewer,

We sincerely appreciate your comprehensive review of our manuscript. We revised the paper in response to your comments and suggestions, and the changes are highlighted in red. Below is our response to your comments:

Comment 1: The authors should add a description of the unit disk graphs, explaining why they can be used to simulate WSNs.

Reply: Thanks for your suggestion! We added references to modeling WSNs with the unit disk graph (UDG) and added a brief description of UDG (lines 24--27; the reference added is: Kuhn, F.; Moscibroda, T.; Wattenhofer, R. Unit disk graph approximation. In Proceedings of the Proceedings of the 2004 joint workshop on Foundations of mobile computing, 2004, pp.).

Comment 2: The updates of gain and loss should be mentioned in the description of TIVC and added in the complexity analysis.

Reply: Thanks for pointing out the issue! We understand the importance of discussing the gain and loss update process, and we added a description of the process of updating gain and loss and considered the process in complexity analysis. The changes made in response to this comment involve lines 171, 172, 181, 182.

Comment 3: The WalkBMS strategy should be briefly described in Related Work.

Reply: Thank you! We added a brief description of the WalkBMS strategy in the "Related Work" section. The changes made in response to this comment involve lines 118--119.

Comment 4: The authors mentioned an interest in working on other graph theory problems on large graphs, a brief description of the challenges that may follow can be added.

Reply: Thanks for your suggestion! Your suggestion of briefly describing challenges that may arise when working on other graph theory problems on large graphs is valuable. We added a related discussion in Section 6 to give readers an overview of the potential future research directions. The changes made in response to this comment involve lines 270--277.

Comment 5: The authors can briefly include in the conclusion a reference to how this work has contributed to solving these practical problems.

Reply: Thank you! In Section 6, we briefly describe how our work contributes to solving practical problems, thereby emphasizing the significance of real-world applications of our algorithms. The changes made in response to this comment involve lines 266--269.

Comment 6:  In line 105, the word "the" needs to be inserted before "age values".

Reply: Thank you for pointing out the missing "the" before "age values" in line 105. We made an amendment to the place. The changes made in response to this comment involve line 100.

Comment 7: The "a" should be added before "tiny perturbation" in the Abstract and Conclusions.

Reply: Thanks! We added the article "a" before "tiny perturbation" in both the Abstract and Conclusions to maintain consistency and accuracy in our presentation. The changes made in response to this comment involve lines 8, 264.

Comment 8: The "age" in line 102 is different from the previous format.

Reply: We apologize for the inconsistency with the term "age" in line 102. We made an amendment to the place.

Once again, we are grateful for your constructive feedback. We are dedicated to addressing these comments thoroughly to enhance the clarity and impact of our paper.

Reviewer 3 Report

This paper is not ready for publication. It is more appropriate for a confrence presentation.

The authors provide a local search for a well known and well studies problem, Maximum Vertex Cover. The main improvement issue of the algorithm provided is called 3-improvments, this means more than one flip search process. The idea of r-flip search is well studied in combinatorial optimization, specifically in binary program. The authors also apply their algorithm in, as they mention, 72 real problems. Given the fact that, there are many studies on MVC and equivalent problems, I believe overall, the paper is not suitable for publication in the sensor journal. (1). The algorithm presented is too simplistic, (2) the authors do not explain well their processes, (3) they do not explain r-flip strategies available in the literature, (4) I am not sure what are these specific benchmark problems they use in the experiment. There are many benchmark problems available on the internet that can be used in experiment. Thus, I suggest the paper should be rejected.

Author Response

Dear Reviewer,

Thank you very much for reviewing our paper on the minimum vertex cover problem (MVC), and thanks for your comments and questions about the paper. We revised the paper in response to your comments and suggestions, where the revised parts are highlighted in red. Below is our response to your comments:

Comment 1: The algorithm presented is too simplistic.

Reply: Thanks for your comments. Local Search has received extensive attention as a heuristic in recent years, not only due to its excellent performance on many intractable problems but also due to its simplicity of principle. In fact, local search algorithms, such as TIVC, have some implementation details that we have not discussed in the previous manuscript, e.g., updating the gain and loss of vertices. We have added more descriptions of our algorithms in the revised version of the paper (lines 171, 172, 181, and 182).

Comment 2: The authors do not explain well their processes.

Reply: Thanks for pointing out the issue! We describe TIVC in more detail in the revised version to ensure that the readers can better understand the procedures and principles. The changes made in response to this comment involve lines 137, 157--160, 168, 169.

Comment 3: They do not explain r-flip strategies available in the literature.

Reply: Thank you very much for this comment. We understand your mentioned r-flip search. However, this is somewhat different from the (j,k)-swap, i.e. k-improvement, used in graph theoretic problems. r-flip attempts to change a portion of the solution to explore the solution space, and r has a wider range of values, which is more applicable to the binary program that you mentioned, whereas the values of j and k for the (j,k)-swap are generally less than 3. This is because too large of a value can significantly perturb the search direction, thus leading to a much lower quality of the searched solutions. In addition, we list some papers solving graph theoretic problems that you may be interested in, none of which describes r-flips.

- Quan, C.; Guo, P. A local search method based on edge age strategy for minimum vertex cover problem in massive graphs. Expert Systems with Applications 2021, 182, 115185.

- Wang, Y.; Cai, S.; Yin, M. Local search for minimum weight dominating set with two-level configuration checking and frequency based scoring function. Journal of Artificial Intelligence Research 2017, 58, 267–295

- Luo, C.; Hoos, H.H.; Cai, S.; Lin, Q.; Zhang, H.; Zhang, D. Local Search with Efficient Automatic Configuration for Minimum Vertex Cover. In Proceedings of the IJCAI, 2019, pp. 1297–13

- Cai, S.; Lin, J.; Luo, C. Finding a small vertex cover in massive sparse graphs: Construct, local search, and preprocess. Journal of Artificial Intelligence Research 2017, 59, 463–494.

Comment 4: I am not sure what are these specific benchmark problems they use in the experiment. There are many benchmark problems available on the internet that can be used in experiment.

Reply: Thanks for the comment. Here we explain in detail the benchmark instances used in our experiments. Each instance is not a problem, but a graph containing n vertices and m edges that are abstracted from various types of real-world networks (e.g., social networks, road transportation networks, biological networks, etc.). The instances we use are well-known instances that are commonly used as tests for graph theoretic problems (e.g., minimum dominating set, maximum independent set, minimum vertex cover, etc.).

We made major revisions throughout the manuscript, all of which are highlighted in red. Thanks again for your review and valuable comments.

Reviewer 4 Report

This work develops a TIVC algorithm to solve the MVC problem. This work is original and interesting, and the authors should be commended for their contribution. The experiment verification of the method shows the result to support the discussion and conclusion of this manuscript.

I have two major suggestions and two minor ones.

Major suggestion 1: the manuscript conducts computational experiments and compares the proposed algorithm with FastVC and EAVC. The results in Table 2 show the advantages of the proposed method in terms of best (strict) solution finding. However, the results also show FastVC and EAVC have 28 and 39 best solution instances, and the proposed method cannot find the best solution in 18/78 instances. The reviewer suggests a deeper discussion (with example) about the limitation of the proposed method, and under what conditions, the proposed method cannot find the best solution.

Major suggestion 2: the reviewer suggests the manuscript add one example of a real-world MVC application, such as a WSN problem, to demonstrate the improvement of the proposed method.

Minor suggestion 1: the manuscript structure needs to be improved. For example, The first paragraph of Section 2 is ‘This section introduces some preliminary knowledge.’. The reviewer suggests the manuscript adds some introduction regarding the overall structure of the corresponding section. For example, ‘This section introduces some preliminary knowledge. Section 2.1 introduces xxx. Section 2.2 discusses xxx’.

Minor suggestion 2: Regarding the future work in Section 6 Conclusions, the manuscript ONLY states ‘In the future, we would like to study the ideas for solving other graph theory problems on large graphs.’. The reviewer suggests a paragraph to discuss future work, for example, how to apply the proposed algorithm to real applications, how to further improve the algorithm, and how to further validate the proposed algorithm.

Writing in standard and accepted level, and can be improved

Author Response

Dear Reviewer,

Thanks for your comprehensive review on the manuscript. We revised the paper according to your comments and suggestions, and those involving changes are highlighted in red. Below is our response to your comments:

Comment 1: The manuscript conducts computational experiments and compares the proposed algorithm with FastVC and EAVC. The results in Table 2 show the advantages of the proposed method in terms of best (strict) solution finding. However, the results also show FastVC and EAVC have 28 and 39 best solution instances, and the proposed method cannot find the best solution in 18/78 instances. The reviewer suggests a deeper discussion (with example) about the limitation of the proposed method, and under what conditions, the proposed method cannot find the best solution.

Reply: Thanks for the suggestion! We understand the importance of discussing the limitations of our proposed algorithm. We have added a description of the limitations of TIVC based on your comments. In addition, in order to more visually show the performance of TIVC on instances of different orders, we added a discussion on this by considering instances of orders  10^4, 10^5, 10^6, and 10^7, respectively (see Section 5.2 in the revised version). The changes made in response to this comment involve lines 237-261. Also, we apologize for the writing error in the paper; there are 72 (not 78) instances in our experiments, and we have corrected it in the revised manuscript.

Comment 2: The reviewer suggests the manuscript add one example of a real-world MVC application, such as a WSN problem, to demonstrate the improvement of the proposed method.

Reply: Thank you very much! We have added an example of a real-world application of MVC, i.e., deploying electronic cameras in city road networks. The changes made in response to this comment involve lines 27--31.

Comment 3: The manuscript structure needs to be improved. For example, the first paragraph of Section 2 is ‘This section introduces some preliminary knowledge.’. The reviewer suggests the manuscript adds some introduction regarding the overall structure of the corresponding section. For example, ‘This section introduces some preliminary knowledge. Section 2.1 introduces xxx. Section 2.2 discusses xxx’.

Reply: Thank you very much for this suggestion to help us optimize our article structure! In the revised version, we divided Section 1 into three subsections, briefly described the subsections of Section 2, added an overview of complexity analysis in Section 4, and briefly described the subsections of Section 5. The changes made in response to this comment involve lines 76, 77, 136, 137, 188--190.

Comment 4: Regarding the future work in Section 6 Conclusions, the manuscript ONLY states ‘In the future, we would like to study the ideas for solving other graph theory problems on large graphs.’. The reviewer suggests a paragraph to discuss future work, for example, how to apply the proposed algorithm to real applications, how to further improve the algorithm, and how to further validate the proposed algorithm.

Reply: Thanks for the suggestion! In the revised manuscript, we added an elaboration of future work in Section 6, including possible problems to be faced, real applications, and general research directions. The changes made in response to this comment involve lines 270--277.

We have checked the paper again, hoping that there are no other mistakes. Thanks again for your review and valuable comments.

Round 2

Reviewer 3 Report

The authors have considered my concerns. I suggest now the paper should be accepted.

Author Response

Dear Reviewer,

Thank you very much for taking the time to review our paper and providing valuable comments on our work!

Reviewer 4 Report

This revision addressed my minor concerns. However, the two major concerns should be addressed as well to meet the standard of publication.

  1. Regarding the discussion of the limitations of the proposed method, line 237-261 just demonstrates and discusses the performance of the proposed method under different orders. However, the reviewer suggests a deep discussion (with examples) on the instances the proposed method cannot find the best result while an alternative method can. For example case ‘wave’ in Table 2. A discussion about under which circumstances the proposed method outperforms alternative methods, while under which circumstances the alternative methods are better is highly suggested.

  2. The reviewer suggests the manuscript adds one example using the proposed method for a real-world WSN problem (in Section 5). NOT JUST A BRIEF DISCUSSION IN INTRODUCTION.

Can be improved

Author Response

Dear Reviewer,

Thank you for taking the time to review our manuscript. We apologize that our previous revisions did not meet your comments, and we have made further revisions to the manuscript based on your comments and suggestions (those involving changes are highlighted in red). Below is our response to your comments:

Comment 1: Regarding the discussion of the limitations of the proposed method, line 237-261 just demonstrates and discusses the performance of the proposed method under different orders. However, the reviewer suggests a deep discussion (with examples) on the instances the proposed method cannot find the best result while an alternative method can. For example case ‘wave’ in Table 2. A discussion about under which circumstances the proposed method outperforms alternative methods, while under which circumstances the alternative methods are better is highly suggested.
Reply: Thanks a lot for the suggestion! We made further improvements to the manuscript based on your suggestions, and we added a discussion of the limitations of TIVC in Section 5.3 (lines 278--290). Furthermore, local search algorithms cannot guarantee their best performance on all instances, which is influenced by the properties of the instances (such as the number of vertices, number of edges, density, average degree, etc.) and parameter settings. We also added the above description to the manuscript. 

Comment 2: The reviewer suggests the manuscript adds one example using the proposed method for a real-world WSN problem (in Section 5). NOT JUST A BRIEF DISCUSSION IN INTRODUCTION.
Reply: Thanks for this suggestion! In Section 5.2, we provided a detailed description of the real-world application of MVC mentioned in Section 1. Using the instance "inf-road-usa" as an example, we explained the practical application of deploying electronic cameras in the U.S. road network (lines 249--264). 

We would like to thank you again for your review and valuable comments. We look forward to continuing our communication with you to improve the quality of the paper.